# Integration of photocatalytic and dark-operating catalytic biomimetic transformations through DNA-based constitutional dynamic networks

Chen Wang [1], Michael P. O'Hagan [1], Ehud Neumann[2], Rachel Nechushtai[2] & Itamar Willner [1✉]

Nucleic acid-based constitutional dynamic networks (CDNs) have recently emerged as versatile tools to control a variety of catalytic processes. A key challenge in the application of these systems is achieving intercommunication between different CDNs to mimic the complex interlinked networks found in cellular biology. In particular, the possibility to interface photochemical 'energy-harvesting' processes with dark-operating 'metabolic' processes, in a similar way to plants, represents an up to now unexplored yet enticing research direction. The present study introduces two CDNs that allow the intercommunication of photocatalytic and dark-operating catalytic functions mediated by environmental components that facilitate the dynamic coupling of the networks. The dynamic feedback-driven intercommunication of the networks is accomplished via information transfer between the two CDNs effected by hairpin fuel strands in the environment of the system, leading to the coupling of the photochemical and dark-operating modules.

[1] Institute of Chemistry, The Minerva Center for Complex Bio-hybrid Systems, The Hebrew University of Jerusalem, Jerusalem, Israel. [2] Institute of Life Science, The Hebrew University of Jerusalem, Jerusalem, Israel. ✉email: itamar.willner@mail.huji.ac.il

The regulation of complex processes by dynamic networks is a defining feature of cellular biology[1–5]. The ability of these biomolecular networks to reconfigure in response to environmental triggers provides a mechanism for the stimuli-driven control of many cellular functions, such as the spatio-temporal regulation of gene expression[6–10]. There is great interest in mimicking such networks using artificial means (systems chemistry) in order to emulate natural processes towards the development of synthetic biomimetic systems[11–15]. In particular, complex networks of interconnected systems synchronized with the environment play important roles in the enhanced fitness and growth vigor of plants, and understanding the dynamic molecular mechanisms regulating plant metabolism, photosynthesis and growth is a continuous scientific challenge[16,17]. Substantial research efforts are directed towards the development of constitutional dynamic networks (CDNs) emulating functions of native network assemblies[12,18–20]. The simplest [2×2] CDN consists of four dynamically interchangeable and equilibrated constituents AA′, AB′, BA′, and BB′. The triggered stabilization of one of the constituents, e.g., AA′ results in the adaptive dynamic reconfiguration of the CDN into a new equilibrated network, where the content of AA′ is enriched at the expense of AB′ and BA′, which each share components with AA′. The dynamic separation of AB′ and BA′ leads to the recombination of B and B′ and the enrichment of BB′. Different molecular and macromolecular CDNs revealing adaptive reconfiguration features triggered by light[21,22], temperature[23], electrical field[24], pH[23], and supramolecular H-bonds[25] were reported, and constitutional dynamic libraries and dynamic reaction networks[26–30] were demonstrated. Recently, we have introduced DNA-based CDNs as a versatile approach for the regulation of a variety of catalytic processes and functional materials[31]. DNA structures are well-suited towards the design of CDNs, as the base sequence comprising the nucleic acid strands provides a rich toolbox to control the stabilization, destabilization, and reconfiguration of the CDN constituents, for example by fuel/anti-fuel strands[32], the $K^+$-ion/crown ether mediated formation/dissociation of G-quadruplexes[33], and the reversible stabilization/destabilization of duplexes by photoisomerizable intercalator units[34]. By engineering the constituents of DNA-based CDNs to perform specific functions, control over their activity may be achieved through the dynamic changes of the composition of the CDN in response to specific triggers. For example, the guided transcription of mRNA and the translation of proteins by a [2×2] CDN was recently achieved[35]. Many biological networks, however, do not operate in isolation but function as part of larger ensembles in which the triggered reconfiguration of one network proceeds with concomitant information transfer to influence the activity of associated networks. By extension, a second network may subsequently regulate the composition of the first, thus enabling a feedback mechanism. Indeed, we have recently demonstrated the feedback-driven regulation of orthogonal enzymatic cascades through the guided intercommunication of two stimuli-responsive CDNs[36]. Beyond the coupling of enzymatic reactions, however, the challenge remains to develop the CDN systems to mimic further biological processes of enhanced complexity. The photosynthetic network in plants is tightly coupled to photorespiration and assimilation networks[37], and to the overall plant metabolic machinery[38,39] and the inter-relationships between the photosynthetic apparatus and the coupled dark networks reveal complex intercommunication patterns[40]. Until now, however, the challenge of achieving the CDN-regulated intercommunication between a photochemical function and a dark-operating function reminiscent of this natural machinery has not yet been addressed. We envisaged that DNA-based CDNs could serve as a means to interlink energy-harvesting and metabolic processes as a step towards emulating the native machinery found in plants.

In the present study, we introduce CDNs as functional modules to control the integration of photochemical and dark-operating processes, mediated through auxiliary fuel strands in the environment of the CDNs. A CDN-guided "photosynthetic" network that drives the light-induced electron transfer (ET) and the photosynthesis of NADPH is coupled to a dark-operating 'metabolic' CDN assembly that stimulates the biocatalytic oxidation of lactate to pyruvate and the cascaded metabolic amination of pyruvate to L-alanine. We demonstrate the intercommunication between the photochemical and dark-operating networks afforded by the environmental components, where the CDN that drives the enhanced photosynthesis of NADPH also interacts with an auxiliary strand to generate the signal that reconfigures the second network accelerating the metabolic process. This CDN-driven enhancement of the metabolic network, in turn, leads to a positive feedback signal to the first network that dictates the complementary acceleration of the photosynthetic process. Thus, interlinking the two networks via the environmental fuel strands provides a means for the dynamic coupling of two distinct biomimetic transformations that emulate the photosynthetic and dark-operating machinery of plant cells.

## Results

**A photosynthetic CDN**. Fig. 1a introduces the dynamic photocatalytic module mimicking photosynthesis, CDN X. The network consists of four constituents where the photosensitizer Zn (II)-protoporphyrin IX (Zn-PPIX) intercalates into the G-quadruplex unit, tethered as photosensitizer to component A of constituent AA′, and the N,N'-dialkyl-4,4'-bipyridinium ($V^{2+}$) electron acceptor is covalently linked to component A′ of constituent AA′ (Supplementary Figs. 1–2). Irradiation of the photocatalytic module results in the effective quenching of the photosensitizer to yield the redox intermediates Zn-PPIX$^+$·/GQ and $V^+$· (Fig. 1a, panel I). In the presence of 1,4-nicotinamide adenine dinucleotide phosphate (NADP$^+$) and ferredoxin-NADP$^+$-reductase (FNR), the formation of reduced cofactor NADPH proceeds, in analogy to the photosystem I. Each of the constituents in CDN includes a loop domain, used to shift the CDN equilibrium through the stabilization of a triplex upon addition of the appropriate trigger strand, panel II. In addition, each of the constituents is engineered to include a different $Mg^{2+}$-dependent DNAzyme reporter unit to cleave fluorophore-quencher ribonucleobase-modified substrates for the quantitative evaluation of the concentrations of constituents, panel III.

Fig. 1a and Supplementary Fig. 3 depict the triggered reconfiguration of CDN X in the presence of auxiliary triggers. Subjecting CDN X to the strand $T_1$ leads to the stabilization of the T-A·T triplex structure in the loop domain, resulting in the stabilization of AA′ and the reconfiguration of CDN X into CDN $X_a$, where AA′ is upregulated, constituents AB′ and BA′ are downregulated and the constituent BB′ is upregulated (Supplementary Figs. 4–6). The reverse displacement of the trigger $T_1$ by the counter trigger $T_1'$ regenerates CDN X. Similarly, treatment of CDN X with the trigger $T_2$ stabilizes the constituent BA′. BA′ and AB′ are upregulated, and AA′ and BB′ are downregulated (Supplementary Fig. 6). The treatment of CDN $X_b$ with the counter trigger $T_2'$ restores CDN X. The quantitative contents of the constituents in different CDNs are shown in Fig. 1b and Supplementary Table 1. The photosensitized ET process proceeding in different CDNs is stimulated by constituent AA′. The absorption spectra of the photogenerated $V^+$· (Fig. 1c) reveal that the $T_1$-upregulated constituent AA′ in CDN $X_a$ leads to the enhanced photoinduced ET, whereas the $T_2$-downregulation of

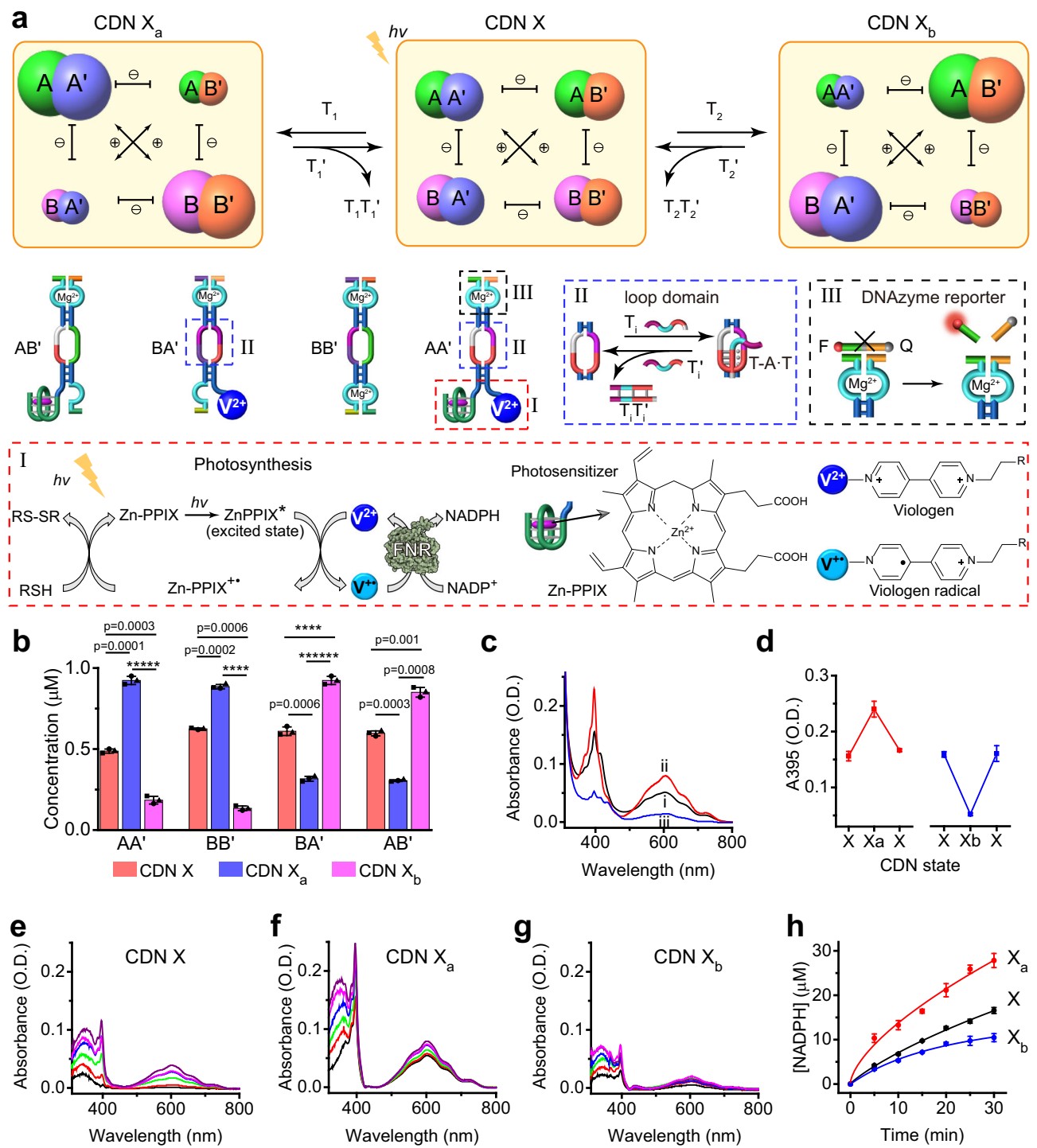

**Fig. 1 Assembly and operation of the photosynthetic constitutional dynamic network (CDN). a** Schematic composition of the 2 × 2 photosynthetic CDN X, its reversibly triggered transition into CDN $X_a$ or CDN $X_b$, and the light-induced reduction of 1,4-nicotinamide adenine dinucleotide phosphate (NADP$^+$) into NADPH by the CDNs. Panel I-The light-induced electron transfer activated by the photosynthetic network, followed by the ferredoxin-NADP$^+$-reductase (FNR)-biocatalyzed synthesis of NADPH, in the presence of thiol electron donor (RSH). Zn-PPIX represents the photosensitizer Zn(II)-protoporphyrin IX and $V^{2+}$ represents N,N'-dialkyl-4,4'-bipyridinium electron acceptor. Panel II-Schematic stabilization of a constituent through the $T_i$-triggered formation of T-A·T triplex and its destabilization by a $T_i'$-induced strand displacement process. Panel III-schematic cleavage of the fluorophore/quencher-functionalized substrate by Mg$^{2+}$-dependent DNAzyme reporter units associated with the constituents. **b** Composition of the constituents in the different reconfigured CDNs. **c** Absorbance spectra corresponding to the bipyridinium radical cation ($V^{+\cdot}$) generated by the photosynthetic network of (i) CDN X, (ii) CDN $X_a$, (iii) CDN $X_b$ (irradiation time interval 30 min). **d** Switchable and reversible formation of the $V^{+\cdot}$, upon the transition CDN X → $X_a$ → X and X → $X_b$ → X. **e–g** Time-dependent absorbance spectra corresponding to the FNR-biocatalyzed photosynthesis of NADPH by CDNs X, $X_a$ and $X_b$, respectively (spectra recorded at time intervals of 5 min for 30 min). **h** Time-dependent photoinduced generation of NADPH by CDNs X, $X_a$, and $X_b$. Error bars in **b**, **d**, **h**, represent mean ± s.d. based on three independent experiments. Two-sided unpaired Student's $t$ test (**b**), ****$p < 0.0001$, *****$p < 0.00001$, ******$p < 0.000001$. $N = 3$ independent experiments were performed. Source data are provided as a Source Data file.

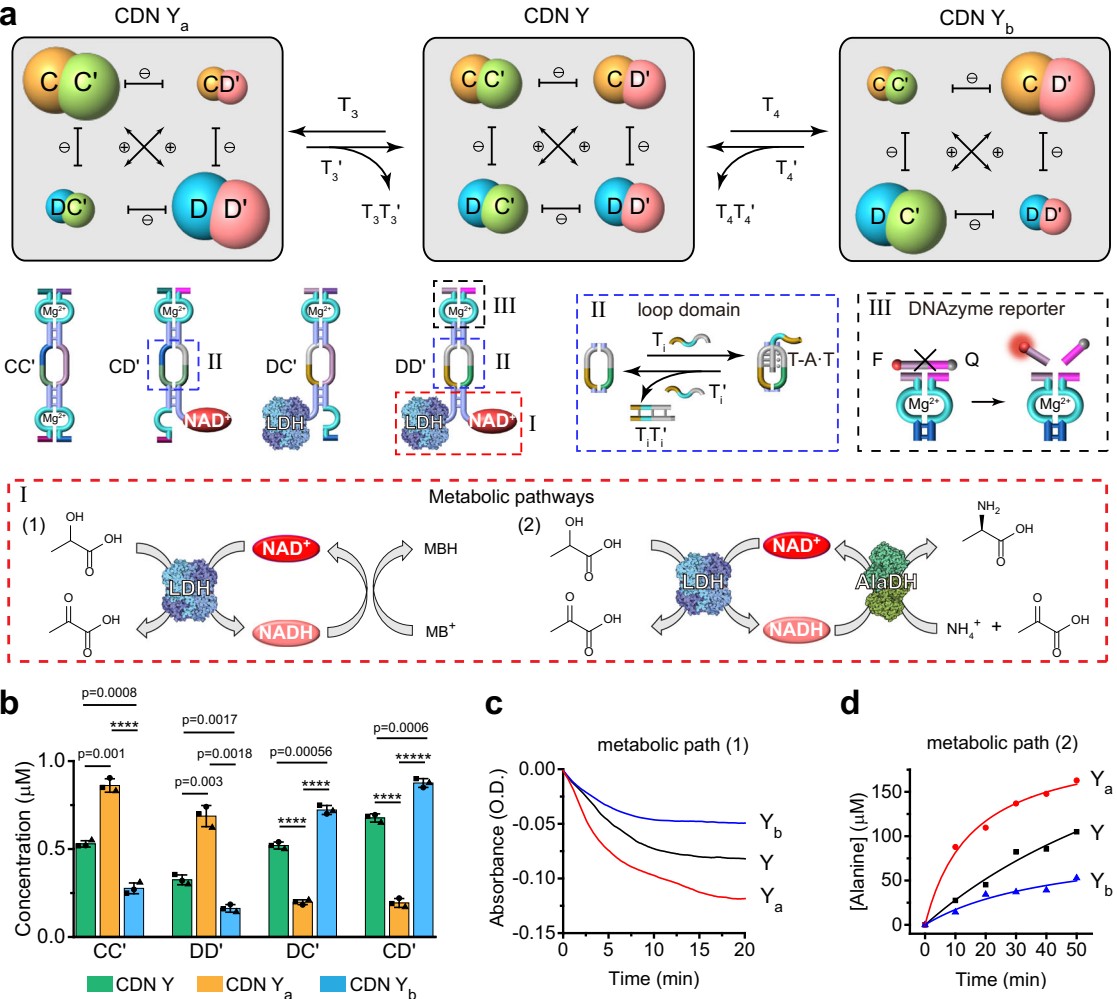

**Fig. 2 Assembly and triggered reconfiguration of the metabolic network. a** Schematic composition of a 2 × 2 CDN Y, its reversibly triggered reconfiguration across constitutional dynamic networks CDNs Y, $Y_a$, and $Y_b$, and the triggered control over metabolic paths. Panel I-Lactate dehydrogenase (LDH)-catalyzed oxidation of lactate to pyruvate and the subsequent reduction of methylene blue ($MB^+$) to MBH as metabolic path (1), and the subsequent alanine dehydrogenase (AlaDH)-metabolic amination of pyruvate to L-alanine, path (2). $NAD^+$ is a cofactor nicotinamide adenine dinucleotide. NADH is a reduced cofactor. Panel II-Reversibly triggered formation/dissociation of T-A·T triplex in the loop domain of a constituent. Panel III-Schematic cleavage of the fluorophore/quencher-functionalized substrate by $Mg^{2+}$-dependent DNAzyme reporter units associated with the constituents. **b** Contents of the constituents in CDNs Y, $Y_a$, and $Y_b$. Error bars represent mean ± s.d. based on three independent experiments. Two-sided unpaired students' *t* test, ****$p < 0.0001$, *****$p < 0.00001$. **c** Time-dependent absorbance changes corresponding to the reduction of $MB^+$ to MBH by the metabolic path (1), in the presence of different CDNs Y, $Y_a$, and $Y_b$. **d** Time-dependent formation of L-alanine by the metabolic path (2) using the different CDNs Y, $Y_a$, and $Y_b$. For duplicate results in **d**, see Supplementary Fig. 15. $N = 3$ (for **b** and **c**) and $N = 2$ (for **d**) independent experiments were performed, respectively. Source data are provided as a Source Data file.

AA′ in CDN $X_b$ reduces the photoinduced ET. Fig. 1d demonstrates the switchable and reversible control over the photoinduced ET process guided by the $T_1$-/$T_2$-triggered reconfiguration of the CDN modules. The secondary CDN-guided synthesis of NADPH driven by the primary photoinduced $V^{+•}$, in the presence of FNR, $NADP^+$, and mercaptoethanol (electron donor), is presented in Fig. 1e–g. The time-dependent build-up of NADPH ($\lambda = 345\,nm$) and $V^{+•}$ ($\lambda = 395\,nm$) is observed. It is evident that the build-up of NADPH by CDN $X_a$ (Fig. 1f) is enhanced as compared with the NADPH generated by CDN X (Fig. 1e), whereas the build-up of NADPH by CDN $X_b$ (Fig. 1g) is reduced as compared to CDN X. Fig. 1h shows the time-dependent formation of NADPH at time intervals of illumination of CDNs X, $X_a$, and $X_b$. The efficiency of production of the photogenerated NADPH is controlled by the efficiency of the primary photoinduced ET process that yields $V^{+•}$.

**A metabolic CDN**. The metabolic CDN module is introduced in Fig. 2 and Supplementary Fig. 7. CDN Y is composed of the constituents CC′, DC′, CD′ and DD′, where the components of DD′ are modified with lactate dehydrogenase (LDH) and $NAD^+$, Supplementary Figs. 2, 8. The metabolic biocatalytic transformation proceeding in CDN Y involves the LDH-biocatalyzed reduction of $NAD^+$ to NADH by lactic acid, and the concomitant formation of pyruvic acid. The biocatalyzed formation of NADH is coupled to the secondary reduction of methylene blue ($MB^+$) to colorless MBH, a process that allows the spectroscopy readout of the time-dependent formation of NADH (Fig. 2a, panel I). In addition, the biocatalyzed formation of NADH and pyruvic acid is coupled to the secondary reductive amination of pyruvic acid, in the presence of $NH_4^+$ and alanine dehydrogenase (AlaDH), to form L-alanine as a metabolic product.

Subjecting CDN Y to trigger $T_3$ stabilizes constituent DD′, resulting in the reconfiguration of CDN Y to CDN $Y_a$, where DD′ is upregulated, DC′ and CD′ are downregulated and CC′ is upregulated (Supplementary Figs. 9–11). The reverse treatment of CDN $Y_a$ with $T_3′$ displaces $T_3$ from DD′ and results in the regeneration of CDN Y. In addition, treatment of CDN Y with $T_4$ stabilizes the constituent CD′, leading to the reconfiguration of CDN Y to CDN $Y_b$, where CD′ and DC′ are upregulated and the constituents CC′ and DD′ are downregulated. Fig. 2b and Supplementary Table 2 show the concentrations of the constituents. The CDNs-guided time-dependent operation of the biocatalytic cascade corresponding to LDH-catalyzed reduction of $NAD^+$ by lactic acid to NADH, and the subsequent reduction of $MB^+$ ($\lambda = 630$ nm) are presented in Fig. 2c and Supplementary Figs. 12–13. The time-dependent reduction of $MB^+$ to MBH is enhanced, in the presence of the $T_3$-triggered reconfigured CDN $Y_a$, and depleted in the presence of the $T_4$-reconfigured CDN $Y_b$, consistent with the upregulation of the constituent DD′ in CDN $Y_a$ and the downregulation of DD′ in CDN $Y_b$, respectively. In addition, Fig. 2d presents the time-dependent CDNs-driven metabolic cascade, where the LDH-biocatalyzed reduction of $NAD^+$ to NADH by lactate is followed by the AlaDH-catalyzed amination of the generated pyruvic acid to yield L-alanine (Supplementary Figs. 14–15, Supplementary Tables 3–5). The rate of formation of L-alanine metabolite is enhanced in the presence of CDN $Y_a$ and dampened by CDN $Y_b$, respectively.

**Intercommunicated photosynthetic and metabolic dynamic networks**. In the next step, efforts to couple the photosynthetic module and the dark-operating metabolic module were undertaken. The principle to intercommunicate the two modules is displayed in Fig. 3a. The constituents BB′ and CC′ in CDNs X and Y were each pre-engineered to include extra $Mg^{2+}$-dependent DNAzyme units. These units are termed activators, integrated into the composite in order to intercommunicate the networks. To intercommunicate the networks, we added two hairpins, $H_a$ or $H_d$ into the mixture of two CDNs. The $H_a$ is designed to be cleaved by the activator in constituent CC′ to yield the fragmented strand $H_{a-1}$ that interacts with constituent AA′ by the stabilization of the triplex T-A·T in the loop domain of AA′ (Fig. 3a route I and Supplementary Fig. 16). This results in the upregulation of AA′ and BB′ and downregulation of AB′ and BA′ (Supplementary Figs. 17–18). That is, the cleavage of $H_a$ by the metabolic module is anticipated to enhance the performance of the photosynthetic module by upregulating AA′. On the other hand, the cleavage of hairpin $H_d$ by the activator of constituent BB′ yields the fragmented strand $H_{d-1}$ that provides an information strand to control the activity of CDN Y (Fig. 3a route II and Supplementary Fig. 19). The binding of $H_{d-1}$ to the loop domain of DD′, and the formation of the T-A·T triplex lead to the stabilization of DD′, the upregulation of DD′ and CC′ and the downregulation of CD′ and DC′ (Supplementary Figs. 20–21). The resulting time-dependent upregulation of DD′ leads, then, to a time-dependent increase in the metabolic performance of CDN Y.

In the first step, the unidirectional intercommunication between the networks using hairpin $H_a$ or $H_d$ was evaluated. Fig. 3b and Supplementary Figs. 22–23 show the formation of NADPH by the photosynthetic module upon exposure to $H_{a-1}$ generated by CDN Y at different time intervals. As the time interval is prolonged, the photosensitized generation of NADPH by the photosynthetic CDN X/FNR is enhanced, consistent with the continuous enrichment of the constituent AA′ by $H_{a-1}$. Fig. 3c depicts the metabolic performance of CDN Y before and after the strand $H_{a-1}$ was supplied to CDN X, by following the reduction of $MB^+$ to MBH. As expected, the metabolic module is unaffected

upon supplying $H_{a-1}$ as a trigger to CDN X. Fig. 3d shows the rate of synthesis of L-alanine by CDN Y upon feeding the two CDNs with $H_{d-1}$ generated at different time intervals (Supplementary Figs. 24–25, Supplementary Tables 6–9). As the time interval of the generation of $H_{d-1}$ is prolonged, the synthesis of L-alanine is enhanced, consistent with the stabilization and time-dependent overexpression of DD′, in the presence of $H_{d-1}$. Fig. 3e shows the spectra of NADPH generated by the photosynthetic module (1-h irradiation) before and after the generation of $H_{d-1}$, implying CDN X is unaffected upon the transfer of the information strand ($H_{d-1}$) to CDN Y. Note that this discussion introduces the positive intercommunication dialog between the CDNs. One may envisage, however, the negative intercommunication between the CDNs. For example, subjecting coupled CDNs to hairpin $H_n$ (cleaved by the activator of BB′) leads to the generation of fragmented product $H_{n-1}$ that stabilizes CD′, the downregulation of DD′ proceeds, resulting in the inhibition of the metabolic module (Supplementary Figs. 26–29). It should be noted that probing the effect of $H_a$ on the metabolic module by means of the methylene blue probe is examined in the dark, without activation of the photosynthetic network, and thus NADPH is not formed and the only source of the methylene blue reducing agent (NADH) originates from the dark LDH-catalyzed oxidation of lactate. Furthermore, upon running the metabolic cycle module composed of lactate/LDH/AlaDH, the equilibrated content of NADH is negligible and does not affect the spectrum of NADPH generated by the photosynthetic module.

Next, we subjected the mixture of the two CDNs to the two hairpins simultaneously. This has a significant effect on the intercommunication between the two CDNs (Fig. 4a). The cleavage of the hairpin $H_a$ yields the strand $H_{a-1}$ that provides the information to upregulate AA′ in the photosynthetic module, leading to the time-dependent increase in the photosensitized ET process and the FNR-catalyzed synthesis of NADPH. However, the upregulation of AA′ is also accompanied by the upregulation of BB′ that leads to the time-dependent enhancement of the cleavage of $H_d$ to form $H_{d-1}$. The latter product provides the information strand to enhance the metabolic module synthesizing L-alanine. The stabilization and upregulation of DD′ are accompanied in turn by the upregulation of CC′ and, thus, increased cleavage of $H_a$ and the enhancement of the photosynthetic module takes place. In the presence of the hairpins $H_a$ and $H_b$, a positive feedback mechanism intercommunicating the CDNs is established (Supplementary Figs. 30–32 and Supplementary Note). Fig. 4b and Supplementary Figs. 33–34 show photosensitized-NADPH generated at time intervals of the feedback-driven intercommunication of the two networks, indicating the generation of NADPH by the photosynthetic module is enhanced. Fig. 4c depicts the rates of the metabolic path (lactate/LDH/$NAD^+$/$MB^+$ cascade) at time intervals of the intercommunication between the networks. As the feedback process enriches AA′ and DD′, the concentration of NADPH is higher and the biocatalytic cascade is enhanced. The results demonstrate a tight relation between the photosynthetic module and the dark-operating module. The upregulation of the photoresponsive constituent of CDN X is accompanied by increased generation of the information transferring strand (via cleavage of $H_d$) that upregulates the metabolic constituent of CDN Y. This leads to a positive feedback mechanism since this reconfiguration of CDN Y, in turn, accelerates the cleavage of $H_a$, which acts as the informational trigger to further enhance the upregulation of the photosynthetic constituent of the first network. It should be noted that the hairpins $H_a$ and $H_d$ that intercommunicate the networks can not be regenerated and they act as fuel strands that control the compositions of the networks while generating degraded waste products.

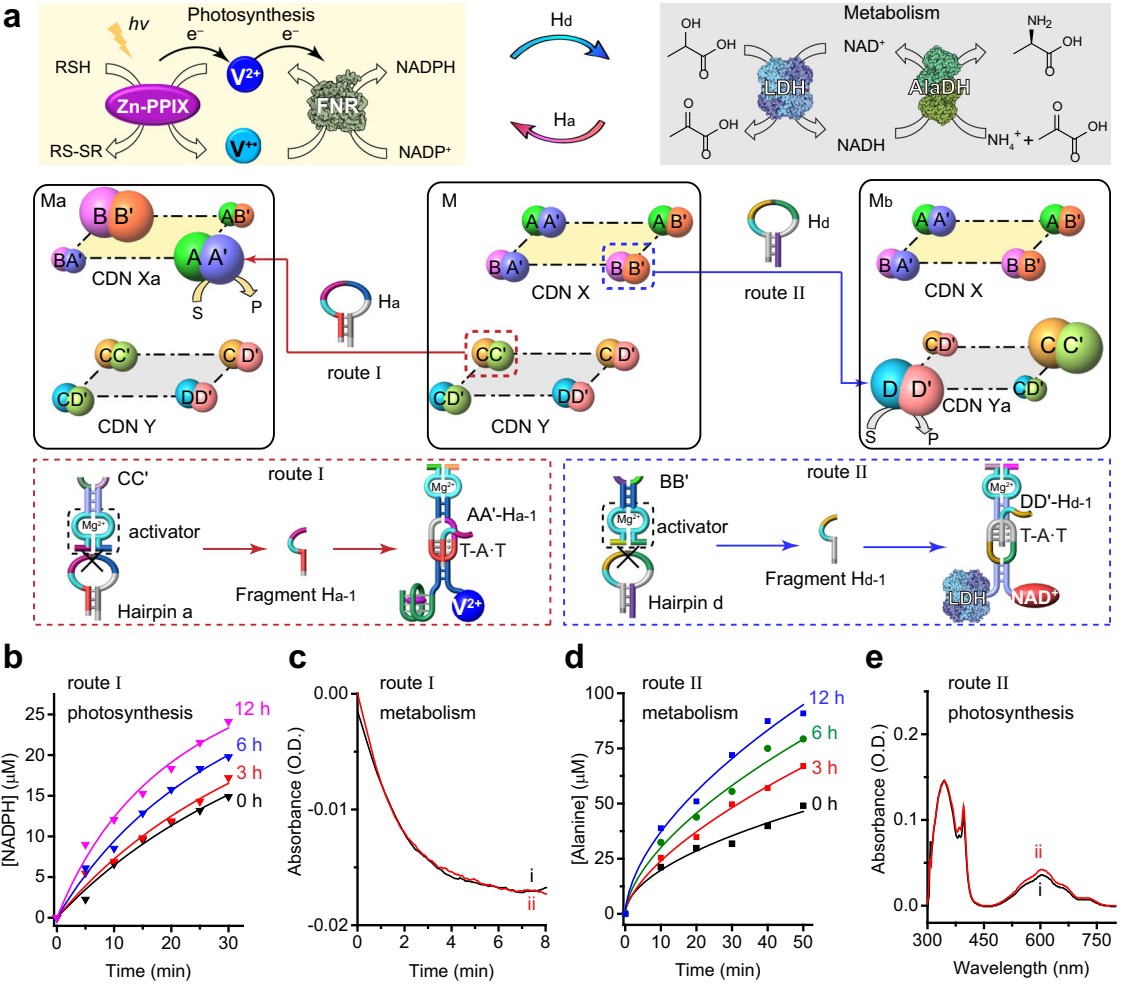

**Fig. 3 Intercommunication between the photosynthetic network (X) and the metabolic network (Y) using hairpin stimulants. a** The mixture M consisting of constitutional dynamic networks CDNs X and Y subjected to the stimulant $H_a$ leads to CDN Y-driven transition of the mixture into state $M_a$ that includes CDN $X_a$ and unaffected CDN Y. The cleavage of hairpin a ($H_a$) by CC′ of CDN Y leads to a fragmented strand $H_{a-1}$ that stabilizes AA′, resulting in the reconfiguration of CDN X into $X_a$ (route I). Subjecting mixture M to hairpin d ($H_d$) leads to the photosynthetic network-guided transition of M into $M_b$ through the cleavage of $H_d$ by BB′ and the generation of a fragmented strand $H_{d-1}$ that stabilizes DD′, while CDN X is unaffected (route II). S and P represent substrate and product, respectively. **b** Time-dependent formation of reduced nicotinamide adenine dinucleotide phosphate (NADPH) by the photosynthetic network at time intervals of treatment of mixture M with the stimulant $H_a$. For duplicate results in **b**, see Supplementary Fig. 23. **c** Time-dependent reduction of methylene blue by the metabolic network: (i) before and (ii) after treatment of mixture M with the stimulant $H_a$. **d** Time-dependent formation of L-alanine by CDN Y at time intervals of treatment of mixture M with the stimulant $H_d$. For duplicate results in **d**, see Supplementary Fig. 25. **e** Absorbance spectra of photosynthesized NADPH by CDN X: (i) before and (ii) after treatment of mixture M with the stimulant $H_d$. $N = 2$ independent experiments were performed. Source data are provided as a Source Data file.

It should be noted, however, that in contrast to natural photosynthesis, where light triggers the intercommunication of the photosynthetic and metabolic modules, the present study applies chemical fuel strands, generated by cleavage of the hairpin structures, as intercommunicating activators. Nonetheless, recent advances in operating dynamic networks demonstrated the light-induced separation of *trans*-azobenzene stabilized nucleic acid duplexes by photoisomerization of *trans*-units into the *cis*-azobenzene state[34]. Such photochemical transformation could mimic the natural system by providing photogenerated fuels that guide the intercommunication of the networks. In addition, the blockage of the photosynthetic module in nature (dark conditions), inhibits autonomously the carbon assimilation process, a feature missing in the present intercommunicated artificial networks, owing to the irreversible formation of the cleaved hairpin fuel strands. This limitation could be resolved, however, by applying photoisomerizable

activator strands, such as the *cis*-azobenzene trigger. The intrinsic dark back isomerization of the *cis*-azobenzene strand to the *trans*-state could then provide an autonomous path to inhibit the metabolic cycle, in analogy to inhibition of the carbon assimilation process under dark conditions of the photosynthetic apparatus.

A further drawback of the present light/dark intercommunicating networks originates from the operation of the coupled CDNs under thermodynamic control. This leads to a permanent background output of the CDNs, and to a limited dynamic range of the switching performance of the coupled networks, far lower than in nature. A strategy to overcome this limitation could involve the activation of the photosynthetic/metabolic networks under out-of-equilibrium, dissipative conditions. Indeed, the operation of dissipative CDNs and of gated networks was recently reported[41,42], providing a potential path to follow.

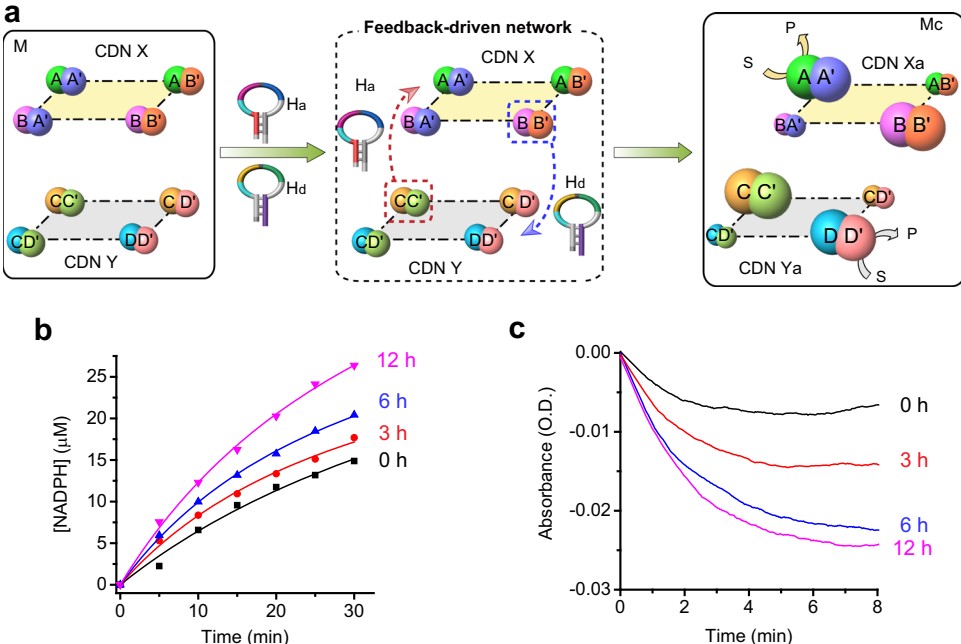

**Fig. 4 Integrated feedback-driven intercommunicating networks X and Y using the mixture of stimulants hairpin a ($H_a$) and hairpin d ($H_d$). a** Subjecting the mixture M to the two stimulants $H_a$ and $H_d$ results in the state $M_c$, where the $H_a$-triggered time-dependent metabolic network (Y)-driven transition of X into $X_a$ and the concomitant $H_d$-triggered time-dependent photosynthetic network (X)-guided transition of constitutional dynamic network CDN Y into $Y_a$ occur. S and P represent substrate and product, respectively. **b** Time-dependent formation of reduced nicotinamide adenine dinucleotide phosphate (NADPH) by the photosynthetic network X at time intervals of activation by stimulant $H_a$-triggered operation of the metabolic network Y. For duplicate results in **b**, see Supplementary Fig. 34. **c** Time-dependent operation of metabolic network (followed by the reduction of methylene blue $MB^+$ to MBH) at time intervals of $H_d$-triggered operation of the photosynthetic network. N = 2 (for **b**) and $N = 3$ (for **c**) independent experiments were performed, respectively. Source data are provided as a Source Data file.

## Discussion

The study introduces two complementary dynamic networks that perform photocatalytic and dark-operating functions, thus adding an additional level of complexity compared to previously reported systems that allow only the control of enzymatic reactions[36]. One network introduces a photosynthetic path, where the control over the light-induced ET and the subsequent catalyzed synthesis of NADPH proceeds, in analogy to the transformation driven by photosystem I. A second dynamic network demonstrates a metabolic path by the input-driven oxidation of lactate and its metabolic transformation to L-alanine. The two networks are intercommunicated, demonstrating the guided activation of the metabolic network by an auxiliary component of the photosynthetic network and the counter control over the photosynthetic network by means of the metabolic network. Finally, the integrated feedback-driven operation of the photosynthetic network and the metabolic network is established by introducing the coupled operation of the two networks, where information transfer between them, mediated by fuel strands in the network environment, exists. Beyond highlighting the integration of the photochemical and dark-operating networks reminiscent of native biological transformations, the study introduces an important path to guide the synthesis of useful materials (L-alanine) by the metabolic network[43]. Beyond advancing System Chemistry by introducing artificial dynamic networks mimicking the photosynthetic/plant metabolic processes, important challenges are ahead of us. These include the photochemical coupling of the photosynthetic and metabolic modules and the development of gated and cascaded, dissipative, out-of-equilibrium photosynthetic/metabolic networks.

## Methods

**Modification of strand A′ with $V^{2+}$.** In all, 60 μL of 0.05 M $V^{2+}$ (10 eq) and 30 μL of 0.01 M sulfo-EMCS (1 eq) were mixed in phosphate-buffered saline (20 mM, pH = 7.24) and incubated at room temperature for 1 h. Then 30 μL of 1 mM strand A′ (0.1 eq) was added and incubated for another 2 h. Excess reactants were removed using Amicon 10 kD cutoff filters. The synthesis route was shown in Supplementary Fig. 2a. The characterization of the modified strand is shown in Supplementary Figs. 35, 36, and Supplementary Table 10.

**Modification of strand D with LDH.** In all, 10 μM of LDH and 1.2 mM SPDP in HEPES buffer (10 mM, pH = 8) were incubated for 1 h. Excess SPDP was removed using Amicon 30 kD cutoff filters. Before modification of strand D with LDH, strand D was treated with TCEP (100-fold excess) for 2 h and washed by using Amicon 3 kD cutoff filters. Next, SPDP-modified LDH was conjugated to strand D (eightfold excess) through a disulfide bond exchange of the activated pyridyldithiol group (see synthetic scheme in Supplementary Fig. 2b). The reaction was performed in HEPES buffer (10 mM, pH = 8) for 2 h. The SPDP coupling efficiency was evaluated by monitoring the increase in absorbance at 343 nm owing to the release of pyridine-2-thione (Supplementary Fig. 10a, b, extinction coefficient: 8080 $M^{-1}$ $cm^{-1}$). Excess DNA was removed using Amicon 30 kD cutoff filters. The enzymatic activity of DNA-modified LDH was ~75% of the activity of the native enzyme (Supplementary Fig. 10c). The DNA labeling ratio of the purified enzyme-DNA conjugates was estimated by measuring the absorbance at 260 and 280 nm (Supplementary Fig. 37 and Supplementary Table 11) and mass analysis was shown in Supplementary Fig. 38.

**Modification of strand D′ with $NAD^+$.** The preparation of D′-$NAD^+$ (see synthetic scheme in Supplementary Fig. 2c) was in the following. In all, 4-carboxyphenylboronic acid (1.8 μL, 50 mM) reacted with EDC (2.3 μL, 10 mg/mL) in 200 μL of MES buffer (10 mM, pH = 5.5) for 5 min, subsequently NHS (3.5 μL, 10 mg/mL) was added and reacted for 10 min. Then, D′ (2.3 μL, 4.4 mM) and $NAD^+$ (3 μL, 5 mM) were added, stirring for 2 h and kept at 4°C overnight. Excess reactants were washed away with Amicon 10 kD cutoff filters.

**Preparation of CDNs.** A sample of 1 mL of CDN (each component 2 μM) was taken as an example to explain the procedure of the preparation of CDNs:

CDN X, including the constituents AA′, BB′, AB′, BA′, was prepared as follows: A (20 μL, 100 μM), A′ (20 μL, 100 μM), B (20 μL, 100 μM), B′ (20 μL, 100 μM) and PPIX (2 μL, 1 mM) were mixed in Tris buffer (10 mM, pH = 7.29) that includes 20 mM $MgCl_2$ and 100 mM $K^+$. The mixture was annealed at 37℃, cooled down to 25℃ at a rate of 0.33℃/min and equilibrated at 25℃ for 12 h. CDN Y, including the constituents LDH/NAD$^+$-DD′, LDH-DC′, CD′-NAD$^+$, CC′ was prepared as follows: LDH-D (20 μL, 100 μM), D′-NAD$^+$ (20 μL, 100 μM), C (20 μL, 100 μM), C′ (20 μL, 100 μM) were mixed in Tris buffer (10 mM, pH = 7.29) that includes 20 mM $MgCl_2$ and 100 mM $K^+$. The mixture was annealed at 37℃ for 1 h, cooled down to 25℃ at a rate of 0.33℃/min, and equilibrated at 25℃ for 12 h.

For the triggered transition of CDN X, triggers $T_1$, $T_1'$ or $T_2$, $T_2'$ are 1.67-fold excess than each component of CDN. After adding triggers into the initial CDN, the final concentration of each component of CDN was 1 μM and the final concentration of trigger was 1.67 μM. The solution was incubated at 28℃ overnight to equilibrate. For the triggered transition of CDN Y, triggers $T_3$, $T_3'$ or $T_4$, $T_4'$ is 2.5-fold excess than each component of CDN. After adding triggers into the initial CDN, the final concentration of each component of CDN was 1 μM and the final concentration of trigger was 2.5 μM. The solution was incubated at 28℃ overnight to equilibrate. After equilibration, the equilibrated CDN (each component 1 μM) was treated with one substrate (5 μM) (sub 1 for AA′, sub 2 for BB′, sub 3 for BA′, sub 4 for AB′, sub 5 for DC′, sub 6 for CD′, sub 7 for CC′, and sub 8 for DD′). The time-dependent fluorescence changes generated by the cleavage of the different substrates by DNAzyme reporter units were measured. By following the rate of formation of the fluorophore-labeled fragment and using appropriate calibration curves of the intact constituent (Supplementary Figs. 4–5 and 9–10), the quantitative evaluation of the concentrations of constituents is achieved.

All the oligonucleotides (including the names and sequences) used in this study were shown in Supplementary Tables 12–15. The base-pairing probabilities were analyzed using an online platform, NUPACK web application (http://www.nupack.org/).

Mass data were collected using Mass Hunter Workstation software 6.00. The data were analyzed using Mass Hunter Quantitative Analysis B.06.00.

**Reporting summary**. Further information on research design is available in the Nature Research Reporting Summary linked to this article.

## Data availability
All data are available in the main text or the supplementary materials and can be obtained from the corresponding author upon reasonable request. Source data are provided with this paper.

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

## Acknowledgements
Our research is supported by the Israel Science Foundation.

## Author contributions
C.W. performed the experiments, analyzed the results, and participated in writing the paper. M.P.O. participated in the scientific analysis of the results and writing the paper. E.N. and R.N. participated in the synthesis of FNR and the analysis of results. I.W. supervised the project and wrote the paper.

## Competing interests
The authors declare no competing interests.
