## [Peer Review File · Nature Communications]

Editorial Note: This manuscript has been previously reviewed at another journal that is not operating a transparent peer review scheme. This document only contains reviewer comments and rebuttal letters for versions considered at *Nature Communications*. Mentions of the other journal have been redacted.

Reviewers' Comments:

Reviewer #1:

Remarks to the Author:

I have already seen this manuscript during submission at [REDACTED]. In my previous report, I asked the authors to add information of the synthesis of DNA conjugates. This point has been properly addressed in the revised manuscript and SI. In my opinion, the manuscript is ready for publication in *Nat. Commun.* The authors might consider to add a remark in the conclusion part that future efforts should be directed to achieve coupling of both CDNs by light irradiation and not only by DNA strands.

Reviewer #2:

Remarks to the Author:

The manuscript has improved a lot. I thank the authors for tuning down their wording and emphasizing the central motivation of their study without overselling the story. Also, in my opinion, all experimental concerns have been addressed by the authors during the revision.

While I am impressed by the complexity of the system, I would still like the authors to comment on three important points in the discussion. First, the dynamic response is not driven by light (the environmental condition), but by the addition of an external compound, which is different from what happens in plants and/or living organisms.

Second, in their system, the first and second network are not coupled with each other, which is of course different to biological systems, where networks dynamically rearrange and reform to perform new functions. In other words: in photosynthesis, when the night sets in, less redox power is provided, which leads ultimately to a shut down of carbon assimilation and triggers the consumption(!) of glucose in the night. The system presented by the authors "only" changes the level of activity upon activation but does not switch and/or revert (in the absence of the signal) completely.

In respect to the latter point, the authors should thirdly also comment on the limitations of their system. The dynamic response of biological systems spans over several orders of magnitude. Here, it is "only" a factor of four, which is achieved by switching from one state to the other state. While this is achieved by an elaborate network, it does still not mimic the full complexity of, which very often allow a complete switch from a very basal activity to full activity.

Re: NCOMMS-21-14359-T

Title: Integration of photocatalytic and dark-operating catalytic biomimetic transformations through DNA-based constitutional dynamic networks

The following point-by-point corrections were introduced into the paper.

Response to reviewers

Reviewer #1

I appreciate the statement of the reviewer that his comments were properly addressed in the revised manuscript.

The reviewer request to “add a remark in the conclusion part that future efforts should be directed to achieve coupling of both CDNs by light irradiation and not only by DNA strands”.

Response: Indeed, this is a valid point that was addressed in the conclusion paragraph by the statement:

“Beyond advancing “System Chemistry” by introducing artificial dynamic networks mimicking the photosynthetic/plant metabolic processes, important challenges are ahead of us. These include the photochemical coupling of the photosynthetic and metabolic modules and the development of gated and cascaded, dissipative, out-of-equilibrium photosynthetic/metabolic networks.”

Furthermore, since the second reviewer addressed the same comment by requesting a discussion on the photochemical coupling of the networks in the main text, we added a dedicated section to the main text explaining how the photochemical coupling of the networks can be eventually achieved, page 10, stating:

“It should be noted, however, that in contrast to natural photosynthesis, where light triggers the intercommunication of the photosynthetic and metabolic modules, the present study applies “chemical” fuel strands, generated by cleavage of the hairpin structures, as intercommunicating activators. Nonetheless, recent advances in operating dynamic networks demonstrated the light-induced separation of *trans*-azobenzene stabilized nucleic acid duplexes by photoisomerization of

trans-units into the *cis*-azobenzene state (*Angew. Chem. Int. Ed.* 2018, 57, 8105–8109). Such photochemical transformation could mimic the natural system by providing photogenerated fuels that guide the intercommunication of the networks.”

Reviewer #2

I appreciate the comment of the reviewer that “the manuscript has improved a lot and the central motivation of the study was presented”.

The reviewer had three very valid comments that are addressed in the corrected paper.

Comment 1: The dynamic response is not driven by light (the environmental condition), but by the addition of an external compound, which is different from what happens in plants and/or living organisms.

Response: Indeed, the reviewer is correct that in nature the coupling between the photosynthetic module and the metabolic module is driven by light rather than using chemical inputs (nucleic acid fuel strands). This issue was addressed in the discussion section, page 10, by adding a dedicated paragraph explaining how photochemical coupling between the modules can be achieved. The added paragraph states:

“It should be noted, however, that in contrast to natural photosynthesis, where light triggers the intercommunication of the photosynthetic and metabolic modules, the present study applies “chemical” fuel strands, generated by cleavage of the hairpin structures, as intercommunicating activators. Nonetheless, recent advances in operating dynamic networks demonstrated the light-induced separation of *trans*-azobenzene stabilized nucleic acid duplexes by photoisomerization of *trans*-units into the *cis*-azobenzene state. Such photochemical transformation could mimic the natural system by providing photogenerated fuels that guide the intercommunication of the networks.”

Comment 2: In photosynthesis, when the night sets in, less redox power is provided, which leads ultimately to a shut down of carbon assimilation and triggers the consumption(!) of glucose in the night. The system presented by the authors "only"

changes the level of activity upon activation but does not switch and/or revert (in the absence of the signal) completely.

Response: The comment that the present networks do not follow the natural principle where the blockage of the photosynthetic module stops the carbon assimilation cycle is certainly correct, and we appreciate clarification of this point by the reviewer.

This comment was explicitly explained and emphasized in the discussion section page 10 and page 11. In fact, we provide a future potential direction to resolve the issue. This comment was addressed and explained as follows:

“The blockage of the photosynthetic module in nature (dark conditions) inhibits autonomously the carbon assimilation process, a feature missing in the present intercommunicated artificial networks, due to the irreversible formation of the cleaved hairpin fuel strands. This limitation could be resolved, however, by applying photoisomerizable “activator” strands, such as the *cis*-azobenzene trigger. The intrinsic “dark” photoisomerization of the *cis*-azobenzene strand to the *trans*-state, could then provide an autonomous path to inhibit the metabolic cycle, in analogy to inhibition of the carbon assimilation process under dark conditions of the photosynthetic apparatus.

Comment 3: The authors should also comment on the limitations of their system. The dynamic response of biological systems spans over several orders of magnitude. Here, it is "only" a factor of four, which is achieved by switching from one state to the other state. While this is achieved by an elaborate network, it does still not mimic the full complexity of, which very often allow a complete switch from a very basal activity to full activity.

Response: Indeed, we agree that the present system reveals limited switching efficiency in contrast to the substantially enhanced switching response of the biological system. We address this comment on page 11, explaining the origin for the present limitation of the dynamic networks and provide a future path to overcome this limitation by dissipative, out-of-equilibrium networks. Appropriate references are included (*J. Am. Chem. Soc.* 2021, 143, 5071–5079; *J. Am. Chem. Soc.* 2020, 142,

17480–17488). The comment was addressed as follows:

“A further drawback of the present light/dark intercommunicating networks originates from the operation of the coupled CDNs under thermodynamic control. This leads to a permanent background outputs of the CDNs, and to a limited dynamic range of the switching performance of the coupled networks, far lower than in nature. A strategy to overcome this limitation could involve the activation of the photosynthetic/metabolic networks under out-of-equilibrium, dissipative conditions. Indeed, the operation of dissipative CDNs and of gated networks were recently reported (*J. Am. Chem. Soc.* 2021, 143, 5071–5079; *J. Am. Chem. Soc.* 2020, 142, 17480–17488), thus providing a path to follow.”

Reviewers' Comments:

Reviewer #2:

Remarks to the Author:

I thank the authors for carefully revising the manuscript and softening, as well as clarifying some of their wording. I congratulate them again for their impressive achievements.